# Managing urban solid waste in Ghana: Perspectives and experiences of municipal waste company managers and supervisors in an urban municipality

Samuel Yaw Lissah[1,2]*, Martin Amogre Ayanore[3], John K. Krugu[4], Matilda Aberese-Ako[5], Robert A. C. Ruiter[1]

1 Department of Work and Social Psychology, Faculty of Psychology and Neuroscience, Maastricht University, Maastricht, The Netherlands, 2 Department of Agro-Enterprise Development, Faculty of Applied Sciences and Technology, Ho Technical University, Ho, Volta Region, Ghana, 3 Centre for Health Policy Advocacy Innovation & Research in Africa (CHPAIR-Africa), Accra, Ghana, 4 KIT Royal Tropical Institute, Amsterdam, The Netherlands, 5 Institute of Health Research, University of Health and Allied Sciences, Ho, Volta Region, Ghana

* slissah@htu.edu.gh

**Data Availability Statement:** The full versions of the transcripts cannot be shared publicly because participants did not consent to it and the limited

## Abstract

Increased population growth and rapid urbanization have resulted in the generation of large quantities of solid waste across major urban cities in Ghana, outstripping local authorities' ability to manage and dispose of waste in a sanitary manner. This study explored the perspectives and experiences of municipal waste company managers and supervisors in the Ho municipality of Ghana on solid waste management practices. A qualitative inquiry was conducted by adopting a phenomenological approach, using in-depth interviews and focus group discussions for data collection. A total of 35 participants, made up of 12 managers and 23 supervisors took part in the study. Transcribed data were imported into NVivo 11.0 software for coding. Content analysis was applied to analyze all transcribed data using processes of induction and deduction. The results showed that organizational capacity, resources, and expertise; community factors such as socio-cultural beliefs and a low sense of responsibility towards solid waste management among urban residents; contextual factors such as regulations, and weak enforcement all influence and shape the level of efficiency and effectiveness of solid waste management practices in the study setting. The findings suggest that policy frameworks and procedures implemented to curb poor urban waste management practices should be systematic and thorough in order to tackle the issue of solid waste in the study setting and Ghana in general. The nature of the identified issues and challenges requires multidimensional and multilevel interventions to provide sustainable solutions for managing urban waste in Ghana.

number of participants being recruited in two easily identifiable settings would threaten the participants' privacy and the confidentiality of their answers. For data inquiries about this research, you can contact or send an email to the administrator from the ethics committee of Ghana Health Service where ethical approval was obtained for the conduct of this study. The details are below: Madam Hannah Frimpong Research & Development Division Ghana Health Service P.O. Box MB 190, Accra Email: hannah.frimpong@ghsmail.org.

**Funding:** The author(s) received no specific funding for this work.

**Competing interests:** The authors have declared that no competing interests exist.

## Introduction

Globally, the volume of solid waste generated is increasing as a result of population density, economic growth, urbanization, and industrialization [1]. It is estimated that an average of 1.9 billion tons of solid waste is generated annually in most cities in the world [2]. Effective solid waste management thus plays a major role in combatting the health and environmental concerns urbanized cities suffer from, particularly in sub-Saharan Africa (SSA) [1, 3, 4]. In SSA, waste generation is estimated to be about 62 million tonnes per year [5]. The effective and efficient management of solid waste is one of the biggest challenges local government authorities face, especially in urban settings [6, 7]. Increased population growth and urbanization have resulted in increased generation of large quantities of solid waste across many cities in developing countries, outstripping local authorities' ability to manage and dispose of solid waste in a sanitary manner [8–10]. Despite spending 30 to 50% of their operational budgets on solid waste management, cities in low- and middle-income countries such as Ghana, only collect between 50 and 80% of the waste generated [11, 12]

In Ghana, about 12,710 tons of solid waste is generated daily, with only 10% collected and disposed of at designated dumping sites [13, 14]. A major challenge in the management of solid waste in Ghana is the collection and disposal process, which are labor-intensive and often not effective. In urban cities in Ghana, issues relating to proper solid waste disposal is a major challenge for the local government authorities. City authorities and waste companies are often overwhelmed by the volume of waste generated daily [15, 16]. The lack of well-planned and efficient strategies to manage waste is one reason for the poor state of solid waste management, particularly by municipal authorities in Ghana [17]. It is estimated that 50 to 70% of the budget of municipal authorities is used to tackle the management and disposal of waste [18]. It has been reported that city authorities in Ghana spend about GHc 6.7 million (US$ 3.45 million) annually on the collection and transport of waste for disposal, and GHc 550,000.00 (US$ 0.28 million) per month to pay waste contractors and for landfill maintenance [19]. Poor sanitation as a result of indiscriminate waste disposal alone is estimated to cost the country $290 million every year- an equivalent to 1.6% of the country's Gross Domestic Product [19].

Major factors hindering proper management of solid waste in Ghana are rapid population growth and urbanization, inadequate supply of waste bins, lack of waste transportation systems, low public awareness on the health consequences of poor waste management, and weak enforcement of environmental regulations [15, 16, 20]. Besides, urban residents' poor behavioral practices towards solid waste is reflected in littering the streets and water passage-ways as well as other public spaces [21]. Low technical know-how on proper solid waste management processes by waste company managers further contribute to the challenges regarding solid waste management [14, 22]. The consequences of indiscriminate or unsafe disposal of solid waste into open drains and water bodies could contribute to flooding and disease outbreaks [21, 23]. Despite successive governments' initiatives such as the empowerment of local government authorities to regulate waste management and policy on private sector participation in waste control, challenges remain in managing waste in many urban cities in Ghana [15, 16]. Other challenges such as inadequate waste infrastructure, inadequate equipment, and insufficient operational funds to support waste management activities have also been reported [13, 15]

Managers and supervisors of waste companies play important roles in waste management processes since they are considered to be key decision-makers regarding waste collection, transportation, and disposal to ensure that it takes place in an effective and efficient manner without contaminating land, air, and water sources [24]. In addition, managers and

supervisors oversee waste management activities such as landfill sites, ensure compliance with existing legislation and bye-laws in the transportation, handling, and disposal of solid waste. They interact with the public on regular basis such as dealing with enquiries and complaints from the general public, as well as investigating and making follow-ups on claims on illegal disposal of solid waste, and work with other regulatory agencies. By these multiple roles, they are in a good position to share experiences on the strengths and weaknesses in solid waste management in the local Ghanian context. Despite the importance of effective waste management for the well-being and health of the general public, which places enormous obligation on managers and supervisors, they are rarely acknowledged. In effect, most skilled workers tend to move to other better sectors, so maintaining sufficient skills and expertise is a problem for waste companies [25, 26]. The work of managers and supervisors is costly, time-consuming, and complex to manage, partially because most of the time they have to work across multiple locations, yet they seem not to be heard. Despite previous studies [27–29] on solid waste management in Ghana, very scarce literature exists till date on the views and perspectives of waste company managers and supervisors on urban waste management practices.

This study seeks to fill this gap in the literature by exploring and understanding how policy implementation, organizational, and community factors influence solid waste management in an urban municipality of Ghana. The findings of the study are also important as the study explored the weight of the responsibility placed on managers and supervisors of waste companies in Ghana, and how they strive to achieve the competing objectives of managing solid waste economically, socially, and in an environmentally friendly manner [30]. The findings will inform local government authorities and other stakeholders on best practices to improve on solid waste management performance in urban cities in Ghana and similar contexts across SSA.

## Materials and methods

### Study design and approval

A phenomenological qualitative research design was used and it employed in-depth interviews (IDIs) and focus group discussions (FGDs) to collect data in the two waste management companies referred to as Company A and Company B. There are only two waste management companies in the study area and the sample came from both companies. In line with the qualitative study approach, the present study seeks to explore and understand human phenomena and experiences of the study participants rather than striving for generalizable findings [31, 32]. The use of phenomenology allows for free expressions of the views and experiences of the participants on the phenomenon being studied [33, 34]. The data used in this study is part of a large qualitative study that examined occupational health risks among domestic waste collectors and their supervisors and managers in the Ho municipality of Ghana. Part of the qualitative evidence involving domestic waste collectors has already been published in a study [35]. For the present study, transcripts on the views and experiences of managers and supervisors of two waste companies on urban solid waste management in the Ho Municipality of Ghana were analyzed and presented.

### Ethical approval

The Institutional Review Board (IRB) of the Ghana Health Service (GHSERC 08/05/17) and the Ethics Review Committee of Psychology and Neuroscience at Maastricht University, the Netherlands (ERCPN 188_10_02_2018) approved the study. The study ensured confidentiality, privacy, and anonymity of both participants and the waste companies. Both written and verbal consent were obtained from each participant prior to data collection.

## Study setting and population

The study was carried out in the Ho Municipality, the administrative capital of the Volta Region of Ghana. In 2018, the projected population of the Municipality was estimated at 213,960, with 51% being females [36]. The Municipality was chosen for the study because of its present challenges in the management of solid waste [37]. Anecdotal evidence in the Ho Municipality shows some reasons for the present challenges of waste management, including poorly built infrastructure and the built-environment arrangements, making it difficult to properly manage waste. In addition, economic activities in the Municipality tend to generate large volumes of solid waste daily, which see minimal collection and disposal regularly. The indiscriminate disposal of solid waste by residents is another reason for the poor state of environmental challenges the Municipality faces [37].

The study population comprised two groups of line managerial staff working in two licensed companies (Company A and B) in the Ho Municipality. Managers and supervisors (participants) were primarily responsible for decision-making, coordination, and supervision of waste collection and disposal in the two companies. To be recruited, eligible participants in companies A and B should have served one year or more in the role as manager or supervisor.

## Sampling procedures and data collection

The data were collected between April-June 2018. Company A is a quasi-private company, while company B is a fully private company. The two companies are responsible for managing public waste collection and disposal in the Ho Municipality. Participants were recruited using purposive sampling techniques. Due to differences in the company's size and staffing numbers, a proportionate sampling technique was used to select participants in the two companies to participate in the study. Company A had a relatively large workforce compared to company B. In company A, 15 supervisors and 10 managers were purposively sampled, while eight supervisors and two managers were sampled purposively from company B to participate in the study (N = 35;12 managers and 23 supervisors). Given that the study employed a qualitative design approach, the 35 study participants were based on thematic saturation rather than a statistical calculation. Therefore, the conclusions are based on analytic generalization and transferability and not on the statistical model of generalizability. It stays the same that wide generalizations would have to rely on further studies as also noted in the discussion section. The different categories of the study participants who were purposively selected enable the study to reached saturation after the field of interest. i.e., no new information on the major themes were obtained [38–40]. The purposive sampling technique supports the investigator to understand and describe in-depth a particular group with similar experiences regarding the challenges of solid waste management in the study setting [41].

Data was collected with the use of In-depth Interviews (IDIs) and Focus Group Discussions (FGDs) guides. The development of the interview guides was informed by an initial review of the literature on issues such as socio-demographic characteristics, infrastructure and capacity of waste companies to deal with practices of solid waste management, attitudes and perceptions of the community towards solid waste management, and barriers (technical and financial) of solid waste management. The interview guides were pre-tested with two managers and two supervisors each from both companies and minor errors in tense structure made for clarity.

Three research assistants (RAs) were recruited to assist in data collection based on their prior experience in field-based data collection and a good understanding of ethical procedures in research. In addition, RAs had a fair knowledge of the study background and were fluent in the native language. The RAs were trained on how to undertake field observations and to

administer informed consent to study participants. The principal investigator supervised and monitored focus group discussions/interviews to ensure that the RAs followed the laid procedures for the data collection by the research team. One RAs conducted the interviews, while another was the recorder and the note taker. In addition, field observations were also documented by the RAs. Before the study, the two solid waste company managers and supervisors were selected with the assistance of the Municipal Environmental Health Officers (MEHOs) in the municipality after informing the MEHOs of the purpose of the study. The managers of the waste companies were then informed by the MEHOs about the study, and were requested to inform their supervisors of the study and its purpose with the research team in attendance. The waste companies were willing to participate in the study and dates for data collection were fixed and agreed upon by the research team and the participants.

## Study instrument

In total, 23 supervisors enrolled in the FGDs. Three FGDs were conducted among the supervisors of the two waste management companies; two FGDs in company A and one FGD in company B. An average of eight participants participated in each FGD session [42]. FGD was useful for the understanding of participants' knowledge and experiences in managing urban waste. FGDs are applied in qualitative research to assess not only what people think but also how they think and why they think that way [43]. Regarding IDIs, 12 interviews were conducted with managers from both waste companies: ten in company A and two in company B at venues of their convenience. FGDs lasted an average of 60 minutes (excluding informed consent process) whilst IDIs lasted for an average of 20 minutes. Interviews were conducted in English until data saturation was reached [39]. Table 1 presents the number of qualitative interviews by type of interview.

## Data analysis

All FGDs and IDIs were audio-recorded and transcribed. The principal investigator (PI) transcribed the interviews while one author (MAA) checked a sample of the transcripts against the audio files to validate the transcriptions. Transcribed data were then imported into QSR NVivo 11.0 software for coding and analysis. Content analysis was applied to analyze all transcribed data using processes of induction and deduction [44, 45]. In the first step of the content analysis, transcripts were read several times to familiarize themselves with the data and to identify common patterns of ideas and views expressed by participants. A follow-up process of 'digging deeper' into the identified lines of transcripts to make meaning of the data (extracting significant views) was undertaken to further create sub-categories and sub-themes. A final inductive stage of analysis was conducted where sub-themes were merged into final themes. The results are presented below as descriptive narratives supported by illustrative quotes.

**Table 1. Type and number of qualitative interviews conducted.**

| Type of interview | Participants | Number (n) |
|---|---|---|
| Focus-Group Discussions | Supervisors company A | 2 |
| | Supervisors company B | 1 |
| | Total FGDs | 3 |
| In-depth Interviews | Managers company A | 10 |
| | Managers company B | 2 |
| | Total IDIs | 12 |

**Table 2. Socio-demographic characteristics of participants.**

| Characteristics | Description | Company A | Company B |
|---|---|---|---|
| Age | 21–30 | 3 | 2 |
| | 31–40 | 12 | 4 |
| | 41–50 | 6 | 2 |
| | 51–60 | 4 | 2 |
| Gender | Female | 3 | 1 |
| | Male | 22 | 9 |
| Years worked in company | 1–5 | 6 | 10 |
| | 6–10 | 17 | - |
| | 11–15 | 2 | - |
| Highest educational level | Senior High School | 15 | 8 |
| | Tertiary | 10 | 2 |
| Job role | Manager | 10 | 2 |
| | Supervisor | 15 | 8 |

## Results

### Socio-demographic characteristics of participants

Twenty-three supervisors and 12 managers were included in the analyses from the two waste companies. The average age of the participants ranged from 31–40 years. The majority of participants were educated to the senior high school level. Further details of participants' socio-demographic characteristics are presented in Table 2.

### The themes that emerged from the study

Three key themes that emerged from the analyses were organizational, community, and contextual factors influencing waste management in the Ho Municipality. Organizational factors that influenced waste management included organizational capacity, resources, and expertise. Community factors included socio-cultural beliefs influencing negative attitudes of residents and a low sense of responsibility towards solid waste management. Contextual factors included rules, regulations, and the enforcement thereof. Fig 1 presents a framework of factors determined in the study to influence waste management practices and the details of the emergent themes are presented in the subsequent sub-sections.

### Organizational factors

**Inadequate solid waste infrastructure and capacity to deal with the process of waste collection.** The managers' and supervisors' who participated in the study avowed that the manual manner in which waste collection and disposal activities are undertaken in the Municipality poses a challenge to the effective and efficient collection and disposal of solid waste. Participants mentioned inadequate equipment to undertake routine waste collection activities, leading to over-reliance on labor-intensive procedures as a common feature of solid waste management processes in the two companies. Study participants reported that solid waste collection trucks, compactors, and other heavy-duty equipment needed for effective solid waste management are mostly inadequate given, the large volumes of waste generated in the Municipality on a regular basis. Participants indicated that occasionally when their trucks break down, large volumes of solid wastes are not collected leading to frequent occurrences of cholera and other dirt-related infectious diseases. Most study participants mentioned the

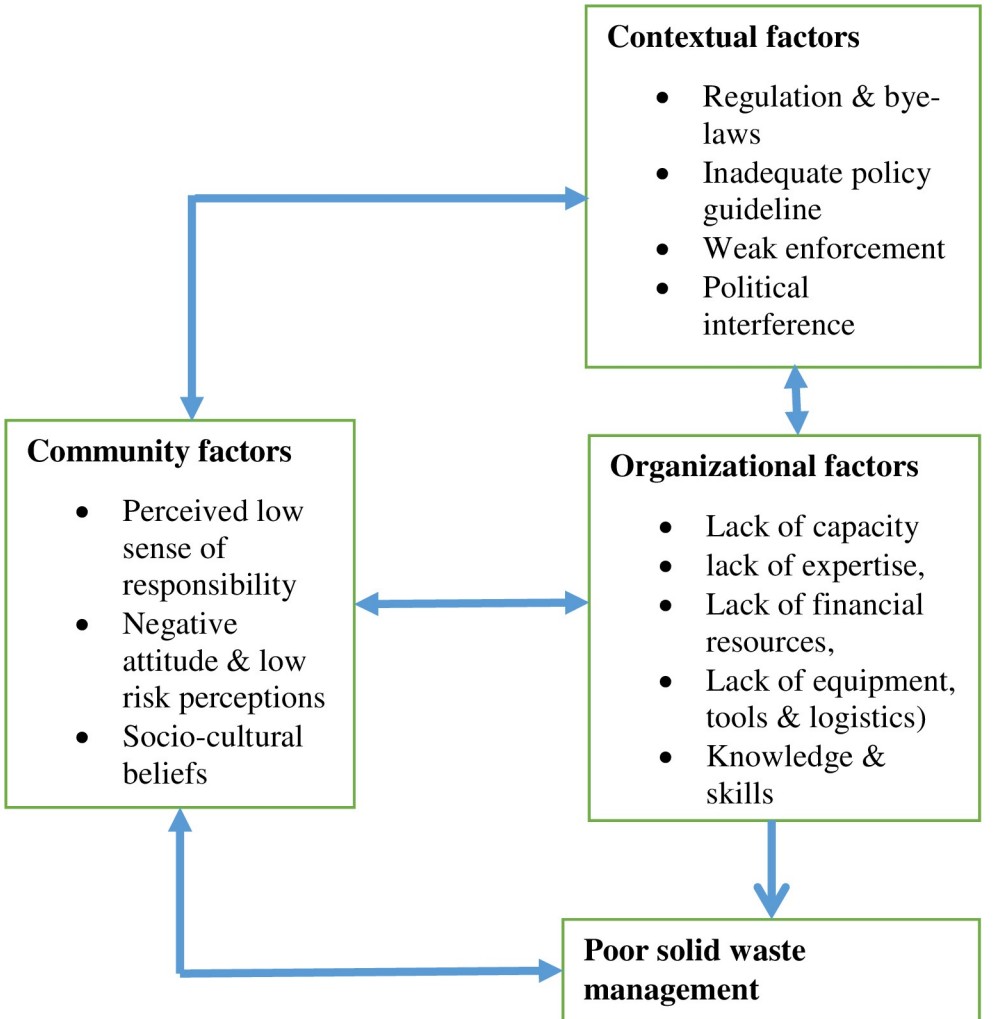

**Fig 1. Framework of factors determined in the study to influence waste management practices.**

insufficient number of public waste bins for temporal storage of waste at some collection points, often leading to indiscriminate disposal of waste by residents.

> . . . *We have solid waste management problems in the municipality, because of the inadequate number of equipment for collecting and disposing of waste (Manager, Company B)*

> . . . *Non-availability of solid waste bins for the urban residents to dump solid waste in the Municipality lead to poor sanitation (FGD Supervisor Company B)*

With regard to the transport of solid waste, participants reported that the common modes of transport for solid waste management in the Municipality is the use of simple tools such as tricycles and wheelbarrows, particularly when financial constraints make it difficult to collect and dispose of large quantities of waste using skip loaders and compactor trucks.

> . . . *we use tricycles, power tillers, three-wheeled tractors, and headloads to collect waste in places by manual sweeping and transporting to dumping sites. (FGD, Supervisor, Company A)*

Participants reported that solid waste disposal methods presently used include landfilling, open burning, and dumping of solid waste at open disposal sites. These methods are often described as "archaic" and not effective. Participants mentioned that incineration, composting, and recycling were some of the best practices in waste management; however, they were not practiced in the Municipality due to lack of expertise, engineering knowledge, and funds. The lack of capacity to adopt some of these best practices for managing solid waste explains the use of "archaic" practices.

> *. . . over here, we are still practicing the archaic method of managing solid waste by landfilling, open dumping, burning, and burying (Manager Company A)*

**Lack of expertise/skilled workforce.** The waste companies are not able to attract personnel with the required skills because of poor conditions of service and remuneration to staff in waste companies. Study participants in both IDIs and FGDs identified inadequate staff with the requisite technical expertise and training to undertake effective supervision and educational support of activities aimed to improve solid waste management processes in the Municipality. They reported that because of their low educational background, they lack the requisite skills/expertise, and capacity to train workers and to support the design of educational strategies to influence residents' positive behavioral practices on solid waste management. The lack of expertise and skilled managers and supervisors is believed to be due to poor remuneration, resulting in the low attraction of skilled workforce to the sector.

> *Ehhh we have problems attracting qualified and dedicated personnel to the sector, because of poor payment of wages and the tediousness of the job (FGD, Supervisor Company B)*

> *Hmm because of our low educational background, only a few of us the managers, and supervisors have knowledge in the proper standard of solid waste management practices. The majority of the managers and supervisors depend on posters or leaflets prepared for solid waste management. (Manager Company B)*

**Financial factors.** The participants mentioned inadequate funds to purchase equipment and to meet other recurrent expenditure needs such as fuel, personal protective equipment (PPE), and emoluments as major impediments to effective solid waste management. It was also reported that the majority of urban residents are not willing to pay for solid waste collection, as they feel that they do not get the needed services. The failure of local government authorities to meet their financial commitments to private waste companies on a timely manner impacts negatively on the sustainability of private waste companies and their ability to undertake frequent waste collection and disposal in the Municipality. One identified bottleneck for the inability of local authorities to timely pay waste companies engaged to manage waste is because of the inadequate collection of sanitation levies from residents and delays from the central government in allocating funds to local authorities to manage waste. Participants reported that they get low returns on investments from sanitation levies although urban residents complained of the high cost of sanitation levies which average GH¢ 300 (USD$ 56) per year.

Participants stated that some urban residents dump solid waste indiscriminately in places such as open drains, and along the streets instead of dumping at appropriate waste collection points that require fee-for-service charges, because of the perceived high cost of sanitation levies.

*. . . the fees we charge are high and most urban residents find it difficult to pay, so they still do illegal dumping of refuse. Anytime it rains, they put the waste into the running water (FGD Company B).*

*. . .how we get funding is that we have a contract with the Municipal Assembly and the Municipal pays for solid waste delivery services. However, we have to pre-finance the solid waste collection before we are paid and the Municipal struggles to pay us and thereby delaying effective solid waste collection (Manager Company A)*

## Perceived low sense of responsibility towards managing solid waste

**Attitudes and risks perceptions of the community towards solid waste.** Negative attitudes of urban residents towards environmental sanitation in general, coupled with the perception that public waste will be collected by "the government" was highlighted by participants as a challenge to waste management in the study setting. The participants attributed these negative attitudes and perceptions by the urban residents to a lack of awareness and education on proper solid waste disposal and the potential effects of poor environmental sanitation on health and wellbeing. Some participants explained that the majority of the urban residents are unaware of the associated risks and harmful effects of improper waste disposal to human life and the environment.

*The challenge is the human attitude towards waste management. Because people dump waste indiscriminately, people will carry refuse from their homes and dump it outside the container, instead of dumping it inside. They claim the waste company will come and collect.'(FGD Company A)*

The participants further indicated that the urban residents hold the views that the local government authority is responsible for the collection, and final disposal of solid waste through their waste management departments, and their Environmental Health and Sanitation Department. Indiscriminate littering is perceived as acceptable by community residents since it offers the opportunity for people to be employed by the waste companies. Managers and supervisors avowed that most residents of the Municipality are not interested and willing to pay for the services of waste collection and disposal provided by waste companies but only feel responsible for the cleanliness of their homes.

*Some of the urban residents will intentionally litter solid waste, and say that if they don't litter solid waste, the waste companies who are responsible for managing solid waste would not have work to do (FGD Company A).*

*The community thinks it is the responsibility of waste management companies to keep the environment clean and will litter indiscriminately (Manager Company B)*

Participants revealed that limited behavior change communication (BCC) activities to sensitize residents on best and safe practices of solid waste management in the Municipality contributed to poor attitudes and low-risk perceptions of the community towards solid waste management. Relating to BCC activities, participants cited inappropriate communication channels when communicating messages on environmental sanitation to residents and limited access to information on safe and hygienic solid waste disposal methods, as attributable reasons why some residents have poor attitudes and perceptions towards solid waste management.

*We announce a way that is not clear when we are in our moving vehicles with our old equipment (Manager Company A)*

*We have poor communication channels and limited access to information to the urban residents for them to support and take part in proper solid waste management (Manager Company B).*

*What we can say is that we do not provide adequate education and awareness raising on proper solid waste disposal to the urban residents, which is also a cause for the challenges of solid waste management (FGD Company B).*

### Environmental/contextual factors to solid waste management

**Contextual factors.** Weak enforcement of environmental rules and bye-laws by local government regulatory agencies was mentioned as a bottleneck to the effective management of solid waste in the Municipality. The majority of the discussants spoke on how legal-political barriers tend to hamper effective solid waste management processes. Managers and supervisors perceived that existing bye-laws were ineffective due to misunderstandings regarding roles and responsibilities among regulatory agencies. The Environmental Health Unit under the Ministry of Local Government feels responsible in part and the Environmental Protection Agency that has the final authority to take action on offenders is located under the Ministry of Environment, Science and Technology. The location of two agencies in different ministries, both responsible for environmental sanitation was noted as creating a weak regulatory framework. Also, respondents revealed that the waste companies hardly conduct effective monitoring and evaluation of routine activities, which hinders their ability to provide support and supervision of staff directly involved in waste collection.

*There is a lack of proper coordination amongst regulatory agencies tackling solid waste and other environmental issues and this affects waste management activities.* (Manager Company A)

*Weak enforcement of existing laws by local authorities tend to hamper our work as implementers in the field of managing waste. Often, we also face political interferences when we try to take legal action against residents or institutions not adhering to state laws on environmental sanitation (Manager Company B)*

**Socio-cultural beliefs influenced community attitudes and practices in waste disposal.** Participants indicated that socio-cultural practices, norms, and beliefs contribute to the negative attitudes of community members towards solid waste management. Another belief held by residents was the feeling of limited there was limited time or space to dispose of waste during the day-time. They reported dumping of waste at night is often preferred since these practices are covertly done. These beliefs resulted in solid waste not being disposed of in a timely and appropriate manner. Thus, piles of solid waste continued to build up in communities.

*The community and some of the residents believe that sweeping is not done in the night since it brings bad luck and poverty to the community (FGD Company B).*

*. . . the lack of involvement of the community to participate and cooperate with the waste companies in solid waste management can lead to indiscriminate solid waste disposal (Manager Company B)*

## Discussion

This study explored Municipal waste company managers' and supervisors' perspectives and experiences on solid waste management in Ghana using a qualitative inquiry approach. Most

of Ghana's discourses on solid waste management have highlighted the apparent breakdown of the regulatory agencies assigned to deal with solid waste management at the local government level [46]. This is based on the fact that poor policy implementation, inadequate funding, and poor technical know-how are some of the factors contributing to the breakdown of regulations on waste management and sanitation [47]. However, this study is unique from existing studies because it provides a description and a good in-depth understanding of how policy implementation, organizational, and community factors influence solid waste management in a developing country context. Although this study focuses on managers and supervisors of waste companies, it is able to bring out how the policy issues, inadequate financial commitment, poor governance systems, and negative socio-cultural norms and practices influence sanitation and solid waste management in the Ghanaian context. The focus of this study in an urbanizing environment in a developing country context is unique in providing relevant findings on how to address the broader goals for sustainable development outlined in the United Nations Sustainable Development Goal 6, targets (to ensure access to safe water resources and sanitation for all by 2030) [48]. Consequently, this study, which is at the heart of solid waste management, is relevant and timely and can help in identifying needs and bottlenecks for future policy development in Ghana and beyond.

The study found that limited infrastructure and capacity to manage solid waste; lack of skilled workforce; perceived low sense of responsibility towards collecting and disposing of solid waste, low-risk perception towards the harmful effects of waste, poor attitudes towards solid waste management in the community, and contextual factors adversely affect the effective management of solid waste in the study setting. Overall, the study findings corroborate existing literature that shows a firm's organizational arrangements, financial and technical resource planning to be essential for implementing effective waste management interventions [13, 49, 50].

The study found that the two waste companies are constrained in dealing effectively with the collection and disposal of solid waste due to the inadequacy of equipment and logistics for waste collection and disposal. In addition, frequent break downs of large heavy-duty equipment and infrastructure disrupt the waste collection value chain, a challenge that impacts negatively on waste management in the municipality. The irregular pattern of waste collection along the waste value chain resulting from inadequate equipment and logistical challenges has been reported in previous studies in Ghana [16, 51]. Aside from the process of waste collection, transport of waste to dumping sites remains a challenge, largely due to the lack of appropriate equipment needed to dispose of large quantities of waste. It was noted that there was over-reliance on simple tools/equipment, particularly tricycles and wheelbarrows to transport and dispose of waste, including large quantities of waste. However, the findings of the study is not consistent with the recommended norm for disposal of large quantities of waste that includes specialized machinery, and equipment such as compactors, skip loaders, and waste disposal trailers [52].

The findings also show that the irregularity in waste collection and disposal creates health and environmental issues in the Municipality, with the occurrences of cholera and other dirt-related infectious diseases as noted by Kogi and Takramah [53]. Health problems are also attributed to the indiscriminate manner in which residents in the municipality dispose of waste, with the excuse that physical infrastructure such as public waste bins for temporal storage of waste at some collection points are inadequate. Evidence shows that unhygienic waste management practices resulted in reported cholera morbidity outcomes in the past in the municipality and other parts of Ghana as well as other settings across SSA [53]. A similar study reported that the absence of effective waste management and recycling systems is causing public health concerns in developing countries. This leads to diseases, hardship, and negative

effects such as loss of income due to illness, and increased health care expenditure for the underprivileged [54]. Experience shows that regular supply of waste bins at various collection points improve easy accessibility for the residents to dump waste [47, 55].

The study also revealed that waste disposal sites are not engineered, leading to solid waste littering in open spaces as a result of using "archaic" practices in waste management. Recommended methods such as recycling and engineered landfills are either absent or rarely used. Instead, archaic practices such as open burning, landfilling, and open dumping of waste are increasingly being used in the Municipality. These archaic practices release toxic organic pollutants such as polycyclic aromatic hydrocarbons, dioxins, and furans into the air, with negative effects on the eco-systems [56]. Our findings suggest poor adherence of the two study waste companies to Ghana's Environmental Sanitation Policy (ESP) on final disposal methods. Ghana's ESP specifically recommends technologies for solid disposal such as sanitary landfill, controlled dumping with cover, incineration, composting, and recycling as standards for the final disposal of urban and large industrial waste [57]. To ensure adherence to the ESP, regulatory agencies such as local authorities in Ghana need to be strengthened to strictly enforce legislation and by-laws on sanitation and solid waste management in the municipality.

Our findings depict a low sense of responsibility among urban residents towards supporting waste management companies' efforts to manage solid waste. The low sense of responsibility is also manifested in negative attitudes and low-risk perceptions of urban residents towards solid waste management in general. These negative attitudes and perceptions by the urban residents could be due to a lack of awareness raising and environmental education on the potential harmful effects of improper solid waste disposal. In this study, the phenomenon of negative attitudes and low-risk perceptions among residents is not new since existing literature shows that local government authorities face similar challenges in dealing with waste management in most urban cities in SSA [58–60].

The study participants perceived that urban residents show strong arguments that the local authority should be solely responsible for matters of waste management, with disregard of citizens' contributions to the process of waste generation and disposal [61, 62] noted that urban residents' perception that generating waste creates job opportunities for domestic waste collectors (DWCs), and waste management companies contributed to the sense of inaction to support local authorities and waste companies to manage waste effectively. These findings suggest the need for a multi-sectoral partnership between public sector agencies to design and implement solid waste management effectively. Behavior change interventions/activities to improve urban residents' awareness, knowledge, and cooperation for managing waste are important [63]. Local authorities and waste companies could identify multiple and innovative channels of targeting and communicating sanitation and hygiene messages to urban residents. Also, evidence indicates that appropriate communication channels can be effective when designing interventions aimed to influence positive sanitation and hygiene practices at the population level [64]

Another important finding from this study suggests that the managers' and supervisors lacked the requisite technical and human resource management skills to effectively manage the waste processing along the value chain, contributing to the culture of poor waste management. Our finding corroborates previous studies that showed low-skilled workforce who play managerial or supervisory roles poses a serious challenge to the effective management of waste [65–67]. The lack of managers and supervisors of the requisite technical skills and capacity could be due to lack of training and poor condition of service in the waste management sector, making skilled workers to shun the job [26, 68, 69].

This study findings also show that urban resident's refusal to pay waste management companies for services provided, contribute to the companies being under-resourced to operate at

optimum. This may explain why the companies are not able to improve on the condition of service of the staff or purchase modern equipment for effective waste management. Our findings confirm previous studies that found inadequate financial resources as one of the major challenges of solid waste management in developing countries [19, 65]. The study also supports previous studies that reported inadequate funding support for managers and supervisors for solid waste management in the urban city councils [68, 70].

The lack of fair and firm enforcement of waste disposal laws by the local authority and political interference contributes to the poor community attitudes towards waste disposal and management. Thus, people litter indiscriminately because they know they would not be prosecuted. Previous studies in Ghana corroborate that the lack of enforcement of waste management legislation is a major obstacle to realizing effective and sustainable waste management [71, 72]. The non-enforcement of the regulations and bye-laws on sanitation may have many underlying reasons including inadequate capacity and financial resources at the local levels [73]. In order to ensure adherence to solid waste policies, there is a need for regulatory agencies such as local authorities to strictly enforce regulations and bye-laws on sanitation and solid waste management in the urban municipality. In addition, the study found that an inefficient or lack of role clarity and coordination among regulatory agencies such as EPA and the Environmental Health Department of the municipal authority affect the implementation of waste control laws in Ghana. Similarly, studies have confirmed that the inefficient or lack of role clarity and coordination amongst the regulatory agencies adversely affect waste management in urban cities in SSA [65]. In resolving the poor solid waste management in the study setting and other developing countries, there is a need for improvements in the coordination, and capacity building among regulatory agencies for effective waste management and environmental sanitation.

Socio-cultural beliefs and practices with regard to indiscriminate solid waste disposal were perceived as contributing to poor community waste disposal practices and cooperation with waste management companies. This is consistent with studies that reported that socio-cultural beliefs such as norms, lack of participation, and cooperation by the communities serve as a challenge to efficient solid waste management in SSA [74, 75]. When urban residents are involved in waste management, they support the waste company managers and supervisors to effectively manage solid waste [1]. The community's participation, and cooperation in solid waste management activities from the planning stage to final disposal, and in structural reforms can enhance and motivate their sense of belonging and ownership [76]. There is a need for managers and supervisors of waste companies to adopt the positive socio-cultural beliefs and cultural practices that would promote sanitation improvement in the study setting and SSA [74]. In addition, there is a need for further research to explore the role of socio-cultural beliefs and cultural practices in solid waste management.

## Study limitation

The findings of this study were limited to two waste companies that operate within the Ho Municipality of the Volta Region of Ghana. Notwithstanding, our results are relevant since managers and supervisors of waste companies in other parts of Ghana may have similar opinions and experiences regarding urban solid waste management. With only the two waste management companies in the study area, it was not practical to get similar participants outside the two companies for the pre-test. Besides, the intention was to include the pre-test data in the study if they were good enough, which was not the case [77]. We acknowledge the potential for bias and false-positive expressions among company managers and supervisors in a bid to conceal vital information regarding waste management processes in their companies. During

data collection, attempts were made to reduce such a bias as possible by in-depth probing, yet one cannot rule out biases. The study design allowed for the use of participatory evaluation to construct and reconstructed participant views during interviews and for understanding the problem under investigation better. The construction and re-construction of views were important to find aberrations and consensus and to minimize false-positive views on the validity of the study findings. This approach enriched the study and provide a strong basis for this study to be generalized across similar Municipal context in Ghana or similar cultures.

## Conclusion

Our study results indicate that waste management is an activity that involves multiple stakeholders such as government, waste companies, and the community playing effective roles. A holistic approach would need to be adopted in intervening because the problems are interlinked. In order to address the challenge multidimensional and multilevel interventions are required. However, further research is necessary for understanding the most appropriate strategy for the involvement of urban residents in solid waste management in poor-resourced settings like Ghana. For effective and efficient solid waste management, the study recommends interventions at the local government, company, and community levels.

The local government authorities need to enforce sanitation and solid waste management by-laws on solid waste management for the waste companies to follow the standard procedure in solid waste management. At the community level, there is a need for the local government authorities to enforce sanitation and waste management regulations in households for the sanitary disposal of waste. The study recommends financial support from the central government to the waste management companies for effective solid waste management practices. There is also a need for training and capacity building for regulatory agencies in order to strengthen them to enable them to strictly enforce legislation and by-laws on sanitation and solid waste management in communities. The local government authorities should provide appropriate and engineered landfill sites for waste companies to dump waste in the appropriate way in order to avoid environmental pollution and flooding.

With regards to the company level, the study recommends that waste companies should follow the standard protocol in waste management. The study also recommends that managers and supervisors of waste companies should intensify awareness raising and education on environmental sensitization to the various target group such as urban residents and the community. In order to promote hygienic way of solid waste management. There is a need for the provision of adequate logistics, solid waste bins, and facilities at vantage points for easy access and dumping of waste. There is a need for waste companies to provide motivation and attractive terms and conditions of service for the staff. This will ensure that waste companies will recruit staff with the requisite skills and qualifications for effective and efficient solid waste delivery service.

At the community level, there is a need to promote community involvement and participation in decision-making processes on sanitation issues in order to help improve on effective solid waste management. In particular, improving waste collection coverage of municipal areas, introducing mass community awareness raising and information campaigns will help to address negative community attitudes towards waste management. Also, it could encourage community members to pay for sanitation levies towards waste management. The implication is that if the various stakeholders follow these recommendations it could lead to effective and efficient solid waste management in the study setting.

## Supporting information

**S1 File.**
(DOC)

## Acknowledgments

We would like to express our profound gratitude to all the study participants for sharing their opinions and experiences with the research team. Our special thanks go to Dr. Samuel Addo, and the research Assistants for assisting us in data collection.

## Author Contributions

**Conceptualization:** Samuel Yaw Lissah.

**Data curation:** Samuel Yaw Lissah.

**Formal analysis:** Samuel Yaw Lissah, Martin Amogre Ayanore.

**Investigation:** Samuel Yaw Lissah.

**Methodology:** Samuel Yaw Lissah, Martin Amogre Ayanore, John K. Krugu, Matilda Aberese-Ako, Robert A. C. Ruiter.

**Project administration:** Samuel Yaw Lissah.

**Supervision:** Martin Amogre Ayanore, John K. Krugu, Matilda Aberese-Ako, Robert A. C. Ruiter.

**Validation:** Samuel Yaw Lissah, Martin Amogre Ayanore, John K. Krugu, Matilda Aberese-Ako, Robert A. C. Ruiter.

**Visualization:** Samuel Yaw Lissah, Robert A. C. Ruiter.

**Writing – original draft:** Samuel Yaw Lissah.

**Writing – review & editing:** Martin Amogre Ayanore, John K. Krugu, Matilda Aberese-Ako, Robert A. C. Ruiter.

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
