## [Decision Letter · Decision Letter 0]

10 Dec 2020

PONE-D-20-33283

Managing urban solid waste in Ghana: Perspective and experiences of municipal waste company managers and supervisors in an urban municipality

PLOS ONE

Dear Dr. Lissah,

Thank you for submitting your manuscript to PLOS ONE. After careful consideration, we feel that it has merit but does not fully meet PLOS ONE’s publication criteria as it currently stands. Therefore, we invite you to submit a revised version of the manuscript that addresses the points raised during the review process.

We look forward to receiving your revised manuscript.

Kind regards,

Bing Xue, Ph.D.

Academic Editor

PLOS ONE

Journal Requirements:

"We acknowledge the Ghana Education Trust (GETFUND) for supporting the first author in pursuing his PhD program."

"The authors received no specific funding for this work."

Reviewers' comments:

Reviewer's Responses to Questions

**Comments to the Author**

1. Is the manuscript technically sound, and do the data support the conclusions?

Reviewer #1: Partly

Reviewer #2: Yes

2. Has the statistical analysis been performed appropriately and rigorously? 

Reviewer #1: Yes

Reviewer #2: Yes

3. Have the authors made all data underlying the findings in their manuscript fully available?

Reviewer #1: Yes

Reviewer #2: Yes

4. Is the manuscript presented in an intelligible fashion and written in standard English?

Reviewer #1: Yes

Reviewer #2: Yes

5. Review Comments to the Author

Reviewer #1: Abstract

The study needs to state the number of possible waste management agencies (sample frame) from which the sample was taken for readers to ascertain if the sample size of two agencies are representative enough to warrant a conclusive results. Clarity should be given on how it is different from existing studies on perspectives of waste management from managers/supervisors to avoid repetition of existing knowledge.

Introduction

The reason provided by authors on why only waste managers/supervisors were sampled for this study is very weak and needs to be strengthened to fit purpose. Otherwise views from waste generators in the communities could be also be obtained to corroborate the information provided by the managers/supervisors.

Materials and Methods

Authors should justify why only two companies were selected for the study, and by provide the calculations they used to obtain 35 respondents whose views provide a conclusive argument for the results. Even though limitations have been provided by the authors as possible bias been introduced, it still does not provide enough backing to justify that the views of 35 respondents may not be enough to provide basis for bringing about multidimensional and multilevel interventions that the authors suggest. The results therefore remain inconclusive and skewed towards the two companies which may be infinitesimal in the business of waste management in the study area, unless proper evidence based justification is provided.

Also, the pre-testing of questionnaires - line 160 - could/should have been done outside the two chosen companies to avoid introduction of errors/biases in ideas for the main questionnaires. Preferably outside the jurisdiction of the study area to make it more reliable.

Results/Discussion

Findings from the study present no new knowledge of what is already known as the challenges of waste management, especially in developing countries. Results should be tailored towards unearthing exciting novel approach on representing the perspectives on waste management that could trigger response from policy makers other than the routine knowledge that yields same ineffective results.

General comments

1. Authors should be consistent in the use of percent and % - refer to lines 58 and 66

2. The study is relevant in the field of waste management. However, findings do not differ so much from existing knowledge and makes it just a repetition without bringing out any exciting information in the field.

Reviewer #2: The study was very well conducted, as demonstrated in the very detailed methodology, which outline the steps taking in arriving at the said results.

The study is pure qualitative and the author outlined in detail the process of sampling, data collection, transcription and analysis, all of which was performed systematically and through a very rigorous process.

The author has also indicated that the data is available upon reasonable request from the first author.

The document is very readable and very well written. No spelling errors identified. And no grammatical errors also observed.

6. PLOS authors have the option to publish the peer review history of their article (what does this mean?). If published, this will include your full peer review and any attached files.

Reviewer #1: No

Reviewer #2: No

---

## [Author Response · Author response to Decision Letter 0]

11 Feb 2021

Detailed revision notes on the revised manuscript

Managing urban solid waste in Ghana: Perspectives and experiences of municipal waste company managers and supervisors in an urban municipality” { Manuscript ID: PONE-D-20-33283}

We thank the two reviewers for their useful feedback and comments on the manuscript. As authors, we feel that the review comments received have helped to improve the manuscript’s quality and readability. In this section, we provide a point-by-point response on the review comments. 

Reviewer #1

Reviewer #1: Comment 1 Abstract

The study needs to state the number of possible waste management agencies (sample frame) from which the sample was taken for readers to ascertain if the sample size of two agencies are representative enough to warrant a conclusive results.

Clarity should be given on how it is different from existing studies on perspectives of waste management from managers/supervisors to avoid repetition of existing knowledge.

Authors’ Response

On the comment regarding sample frame, the number of possible waste management agencies from which the sample was taken and its representativeness are provided on page 6, lines 120-124, and also on page 8, lines 163-170 of the revised manuscript.

On page 20, lines 406-423 of the revised manuscript, the authors have provided further and detailed reasons that explain how this manuscript and the perspectives of managers/supervisors differ and is unique from other previous studies in Ghana. 

Reviewer #1: Comment 2

Introduction

The reason provided by authors on why only waste managers/supervisors were sampled for this study is very weak and needs to be strengthened to fit purpose. Otherwise, views from waste generators in the communities could be also be obtained to corroborate the information provided by the managers/supervisors.

Authors’ Response

In the revised manuscript (page 4-5, lines 87-97), the authors have provided reasons why managers/supervisors were sampled for this study. This study was designed to address specific research gaps in relation to company managers/supervisors' contribution to solid waste management in Ghana. No data was collected at the community level among waste generators. 

Reviewer #1: Comment 3

Materials and Methods

Authors should justify why only two companies were selected for the study, and by provide the calculations they used to obtain 35 respondents whose views provide a conclusive argument for the results. Even though limitations have been provided by the authors as possible bias been introduced, it still does not provide enough backing to justify that the views of 35 respondents may not be enough to provide basis for bringing about multidimensional and multilevel interventions that the authors suggest. The results therefore remain inconclusive and skewed towards the two companies which may be infinitesimal in the business of waste management in the study area, unless proper evidence based justification is provided.

Authors’ Response

The justification for which only two waste companies were selected has been addressed. See page 6, lines 120-124, and also page 8, lines 163-170 of the revised manuscript 

Also, the pre-testing of questionnaires - line 160 - could/should have been done outside the two chosen companies to avoid introduction of errors/biases in ideas for the main questionnaires. Preferably outside the jurisdiction of the study area to make it more reliable.

 Authors’ Response

This has been addressed on page 26 lines 547-550 understudy limitations.

Reviewer #1: Comment 4

Results/Discussion

Findings from the study present no new knowledge of what is already known as the challenges of waste management, especially in developing countries. Results should be tailored towards unearthing exciting novel approach on representing the perspectives on waste management that could trigger a response from policy makers other than the routine knowledge that yields same ineffective results.

Authors’ Response

This study has unearthed context-specific perspectives of how managers/supervisors view the societal challenge of solid was management in Ghana. The findings add to existing evidence on the subject of waste management and the institutional and societal bottlenecks associated with waste management in Ghana. In the revised manuscript (page 5-6 lines 107-115), the authors have provided key results that this study unearthed and how these findings can shape the course of waste management policies in Ghana. 

Reviewer #1: Comment 5

General comments

1. Authors should be consistent in the use of percent and % - refer to lines 58 and 66

Authors’ Response

The text has been revised to ensure consistency in the use of percent and % ( see page 3 line 65) as recommended.

2. The study is relevant in the field of waste management. However, findings do not differ so much from existing knowledge and makes it just a repetition without bringing out any exciting information in the field.

Authors’ Response

See response to comment 1 above on the added value of the study on page 20 lines 406-423. 

Reviewer #2

Reviewer #2; Comment: The study was very well conducted, as demonstrated in the very detailed methodology, which outline the steps taking in arriving at the said results.

The study is pure qualitative and the author outlined in detail the process of sampling, data collection, transcription and analysis, all of which was performed systematically and through a very rigorous process.

The author has also indicated that the data is available upon reasonable request from the first author.

The document is very readable and very well written. No spelling errors identified. And no grammatical errors also observed.

Authors’ Response

We thank the reviewer for the feedback and comments on the manuscript. The authors are very grateful for these positive comments.

---

## [Decision Letter · Decision Letter 1]

26 Feb 2021

Managing urban solid waste in Ghana: Perspectives and experiences of municipal waste company managers and supervisors in an urban municipality

PONE-D-20-33283R1

Dear Dr. Lissah,

We’re pleased to inform you that your manuscript has been judged scientifically suitable for publication and will be formally accepted for publication once it meets all outstanding technical requirements.

Kind regards,

Bing Xue, Ph.D.

Academic Editor

PLOS ONE

Additional Editor Comments (optional):

Reviewers' comments:

Reviewer's Responses to Questions

**Comments to the Author**

1. If the authors have adequately addressed your comments raised in a previous round of review and you feel that this manuscript is now acceptable for publication, you may indicate that here to bypass the “Comments to the Author” section, enter your conflict of interest statement in the “Confidential to Editor” section, and submit your "Accept" recommendation.

Reviewer #1: All comments have been addressed

2. Is the manuscript technically sound, and do the data support the conclusions?

Reviewer #1: Yes

3. Has the statistical analysis been performed appropriately and rigorously? 

Reviewer #1: Yes

4. Have the authors made all data underlying the findings in their manuscript fully available?

Reviewer #1: Yes

5. Is the manuscript presented in an intelligible fashion and written in standard English?

Reviewer #1: Yes

6. Review Comments to the Author

Reviewer #1: The authors have sufficiently addressed arising issues from the previous review and have satisfactorily provided evidence in the revised format to back their conclusion.

Where few sample size were used, authors have justified the reasons which are sound to merit scientific publication.

7. PLOS authors have the option to publish the peer review history of their article (what does this mean?). If published, this will include your full peer review and any attached files.

Reviewer #1: No

---

## [Editor Report · Acceptance letter]

2 Mar 2021

PONE-D-20-33283R1 

Managing urban solid waste in Ghana: Perspectives and experiences of municipal waste company managers and supervisors in an urban municipality 

Dear Dr. Lissah:

I'm pleased to inform you that your manuscript has been deemed suitable for publication in PLOS ONE. Congratulations! Your manuscript is now with our production department. 

Kind regards, 

on behalf of

Professor Bing Xue 

Academic Editor

PLOS ONE